# Safety and Immunogenicity of the Intranasal Vaccine Candidate Mambisa and the Intramuscular Vaccine Abdala Used as Booster Doses for COVID-19 Convalescents: A Randomized Phase 1–2 Clinical Trial

**DOI:** 10.3390/vaccines12091001

**Published:** 2024-09-01

**Authors:** Gilda Lemos-Pérez, Yinet Barrese-Pérez, Yahima Chacón-Quintero, Rolando Uranga-Piña, Yisel Avila-Albuerne, Iglermis Figueroa-García, Osaida Calderín-Marín, Martha M. Gómez-Vázquez, Marjoris Piñera-Martínez, Sheila Chávez-Valdés, Ricardo Martínez-Rosales, Lismary Ávila-Díaz, Amalia Vázquez-Arteaga, Hany González-Formental, Giselle Freyre-Corrales, Edelgis Coizeau-Rodríguez, Miladys Limonta-Fernández, Marta Ayala-Avila, Eduardo Martínez-Díaz, Eulogio Pimentel-Vazquez, Gerardo Guillen

**Affiliations:** 1Center for Genetic Engineering and Biotechnology (CIGB), P.O. Box 6162, La Habana 10600, Cuba; yahima.chacon@cigb.edu.cu (Y.C.-Q.); sheila.chavez@cigb.edu.cu (S.C.-V.); ricardo.martinez@cigb.edu.cu (R.M.-R.); lismary.avila@cigb.edu.cu (L.Á.-D.); amalia.vazquez@cigb.edu.cu (A.V.-A.); hany.gonzalez@cigb.edu.cu (H.G.-F.); giselle.freyre@cigb.edu.cu (G.F.-C.); edelgis.coizeau@cigb.edu.cu (E.C.-R.); miladys.limonta@cigb.edu.cu (M.L.-F.); marta.ayala@cigb.edu.cu (M.A.-A.); 2National Coordinating Center for Clinical Trials (CENCEC), La Habana 11300, Cuba; yinet@cencec.sld.cu (Y.B.-P.); rolando@cencec.sld.cu (R.U.-P.); yisel@cencec.sld.cu (Y.A.-A.); 3Hermanos Ameijeiras Clinical-Surgical Hospital, La Habana 10400, Cuba; iglermis@infomed.sld.cu; 4Manuel Ascunce Domenech Provincial Clinical-Surgical Teaching Hospital, Camagüey 70100, Cuba; cmosaida.cmw@infomed.sld.cu; 5Pedro Raúl Sánchez Teaching Polyclinic, Pinar del Río 20100, Cuba; dirpolrs@princesa.pri.sld.cu; 6Saturnino Lora Provincial Clinical-Surgical Hospital, Santiago de Cuba 90100, Cuba; marjorisp@infomed.sld; 7Ministry of Science, Technology and Environment (CITMA), La Habana 10400, Cuba; eduardo@citma.gob.cu; 8Biotechnology and Pharmaceutical Industries Group (BioCubaFarma), La Habana 10800, Cuba; eulogio@oc.biocubafarma.cu; 9Latin American School of Medicine (ELAM), La Habana 19108, Cuba

**Keywords:** SARS-CoV-2, COVID-19 vaccines, convalescents, booster, intranasal, intramuscular, vaccination, Cuba

## Abstract

A phase 1–2, prospective, multicenter, randomized, open-label clinical trial (Code RPCEC00000382), with parallel groups, involving 1161 participants, was designed to assess the safety and immunogenicity of two Cuban COVID-19 vaccines (Mambisa and Abdala) in boosting COVID-19 immunity of convalescent adults after receiving one dose of either vaccine. The main safety outcome was severe vaccination adverse events occurring in <5% of vaccinees. Main immunogenicity success endpoints were a ≥4-fold anti-RBD IgG seroconversion or a ≥20% increase in ACE2-RBD inhibitory antibodies in >55% of vaccinees in Phase 1 and >70% in Phase 2. Neutralizing antibody titers against SARS-CoV-2 variants were evaluated. Both vaccines were safe—no deaths or severe adverse events occurred. Mild intensity adverse events were the most frequent (>73%); headaches predominated for both vaccines. Phase 1 responders were 83.3% (*p* = 0.0018) for Abdala. Mambisa showed similar results. Phase 2 responders were 88.6% for Abdala (*p* < 0.0001) and 74.2% for Mambisa (*p* = 0.0412). In both phases, anti-RBD IgG titers, inhibition percentages and neutralizing antibody titers increased significantly after the booster dose. Both vaccines were safe and their immunogenicity surpassed the study endpoints.

## 1. Introduction

Since the WHO first characterized COVID-19 as a pandemic on 11 March 2020 [1], researchers have developed various medications and vaccines to improve therapies and its control.

SARS-CoV-2 infection provides natural immunity due to the virally-induced humoral and cellular memory response, but both disease- and vaccine-induced immunity tend to wane over time [2,3]. As the Omicron variant (B.1.1.529) and its subvariants have emerged globally and spread rapidly, a high risk of breakthrough infection and reinfection has been observed in vaccinees and those previously infected [4], hence the necessity for vaccine booster doses. Recent studies suggest that hybrid immunity (natural plus vaccination) is longer lasting and more effective than disease-induced immunity or vaccination alone [5], emphasizing the importance of vaccinating previously infected individuals.

In Cuba, five COVID-19 vaccines have been developed [6]. Abdala vaccine and Mambisa vaccine candidates (henceforth Abdala and Mambisa) are obtained and manufactured at Havana’s Center for Genetic Engineering and Biotechnology (CIGB). Both are recombinant protein subunit vaccines based on the receptor binding domain (RBD) fragment of the SARS-CoV-2 spike protein S produced in *Pichia pastoris* yeast (now *Komagataella phaffii*) [7]. In Abdala, RBD is adjuvanted with aluminum hydroxide gel and injected intramuscularly. Mambisa’s formulation includes RBD plus the hepatitis B virus core antigen (HBcAg) expressed in *Escherichia coli*; it does not contain adjuvant and its application is intranasal. The rationale of this nasal vaccine is based on the SARS-CoV-2 transmission route due to its respiratory tropism. Mambisa’s benefits include antigen delivery at the infection site, elicitation of respiratory tract mucosal immunity and needle-free administration.

Previous clinical trials for Abdala and Mambisa assessed safety and immunogenicity in three-dose schedules in seronegative adults [8,9], and also as a booster dose in healthy adults [10]. Abdala was the first COVID-19 vaccine approved in Cuba for emergency use [11] and was a turning point for epidemic control in the country [12,13]. Studies with Abdala have also been conducted in children, adolescents and pregnant women [14,15].

The objective of this clinical trial was to assess the safety and immunogenicity of Abdala and Mambisa vaccines when administered as booster shots to individuals who have recovered from COVID-19. This is the first report on the immune response of convalescent individuals to SARS-CoV-2 after receiving a booster dose of the Abdala or Mambisa vaccines. Furthermore, it is also the first study to document the use of a nasal subunit vaccine.

## 2. Materials and Methods

### 2.1. Vaccines

Abdala contains 50 µg RBD per 0.5 mL [11], and is administered in the deltoid region. Mambisa contains 50 µg RBD plus 40 µg HBcAg per 0.2 mL (0.1 mL per nostril). The RBD antigen sequence is identical to the ancestral SARS-CoV-2 Wuhan-Hu-1 strain (NCBI Acc. No. YP_009724390). Both products were stored at 2–8 °C, as recommended. Mambisa was approved by the Center for State Control of Medicines, Equipment and Medical Devices (CECMED) for this study [16].

### 2.2. Nasal Devices for Vaccine Delivery

Three nasal devices were used for Mambisa administration: an intranasal atomization device or atomizer (a prototype employed in Phase 1 [CNEURO, Havana, Cuba] similarly designed as the commercial device employed in Phase 2 [Wuxi NEST Biotechnology, Wuxi, China]) and identified as AZ, a nasal spray (Gaasch Packaging, Glasgow, UK) identified as S, and a nasal dropper (Sopac Medical, Goussainville, France) identified as D (see Appendix A for details).

### 2.3. Trial Design

A phase 1–2, prospective, multicenter, randomized, open-label clinical trial with parallel groups was designed to assess safety and immunogenicity of Mambisa and Abdala in boosting COVID-19 immunity of convalescent adults after receiving a single dose of either vaccine.

Phase 1 was an exploratory study to select (in terms of safety and immunogenicity) the appropriate nasal device for Mambisa administration. The selected device was used in Phase 2 to confirm and reinforce Phase 1 results.

Figure 1 shows the study flow diagram. In Phase 1, enrollment of 120 participants was completed at the Hermanos Ameijeiras Clinical-Surgical Hospital in Havana, Cuba (Figure 1A). Phase 2 included 1041 participants (Figure 1B) enrolled at Hermanos Ameijeiras Clinical-Surgical Hospital and at three more clinical sites in other cities: Manuel Ascunce Domenech Provincial Clinical-Surgical Teaching Hospital (Camagüey), Pedro Raúl Sánchez Teaching Polyclinic (Pinar del Río) and Saturnino Lora Provincial Clinical-Surgical Hospital (Santiago de Cuba).

The trial was conducted in certified vaccination areas at each clinical site, and participants were vaccinated by nurses certified for this procedure. Monitors verified this process at each site, as well as the accuracy of case report forms (CRF) and good clinical practice (GCP) procedures.

This study was registered in the Cuban Public Registry of Clinical Trials (Code RPCEC00000382, https://rpcec.sld.cu/trials/RPCEC00000382-En/, accessed on 25 May 2024).

### 2.4. Participants

Inclusion criteria: Subjects aged 19–80 years, COVID-19 convalescents, at least two months after recovery, healthy or with compensated comorbidities. All convalescents had been infected with SARS-CoV-2 before November 2021.

Exclusion criteria: listed in Figure 1 and detailed in Appendix A.

### 2.5. Variables

Safety. Adverse events (AE) type (local or systemic), duration, severity and causal relationship. Severity was classified as not severe or severe (when hospitalization or its prolongation was required, or the reaction was life-threatening or contributed to a patient’s death). AE intensity was established in three levels: (a) mild, vaccination well tolerated, caused minimal discomfort and did not interfere with daily activities; (b) moderate, annoying enough to interfere with daily activities; and (c) severe, when it interrupted daily activities.

Immunogenicity. Increase of serum IgG and IgA antibody titer against RBD; percentage inhibition of RBD-ACE2 (angiotensin-converting enzyme 2) binding; and neutralizing antibody (NAb) levels for live SARS-CoV-2 variants.

### 2.6. Sample Size

For Phase 1, four 30-person experimental groups (120 in total) were established. This exploratory phase did not necessitate controlling the significance level or power. In Phase 2, the number of subjects was a function of the number of individuals who met the main success criterion of Phase 1. Considering 10% dropout, *N* = 232 subjects were calculated for each of the four strata provided (Mambisa, age ≤ 60; Mambisa, age > 60; Abdala, age ≤ 60; Abdala, age > 60), making it necessary to recruit 928 subjects. The significance level defined (α = 0.0125) guaranteed a global significance level α = 0.05 and a power of 85% to conduct the four hypothesis tests, which constituted the main objectives for analysis.

### 2.7. Randomization

In both study phases, two control variables were taken into account for participants’ randomization: age and clinical category of COVID-19 infection. Age was stratified into two groups: ≤60 and >60 years. The clinical category of COVID-19 disease was stratified by symptoms (symptomatic or asymptomatic) and severity of symptoms (mild, moderate, or severe).

In Phase 1, subjects were randomly distributed into four groups. Three were assigned to Mambisa, differentiated according to the device used for IN delivery. The fourth group received Abdala (Figure 1A).

In Phase 2, subjects were randomized in two groups for the Abdala and Mambisa vaccinations (Figure 1B). Enrollment/allocation was made after the main researcher verified compliance with the subject selection and exclusion criteria.

Randomization lists for both phases were prepared by the trial’s statistical administrator using the Random Allocation Software for Windows (v 1.2).

### 2.8. Intervention

The Abdala group received a booster dose of 0.5 mL intramuscularly (deltoid region). Those in the Mambisa group received a booster dose of 200 µL (100 µL per nostril) via the corresponding nasal application device. For intranasal administration, each participant was seated with the head slightly tilted back, with support throughout the procedure. The product was applied into the first nostril after exhaling, maintaining the head position for one minute; the same procedure was repeated in the other nostril. The Mambisa vaccine application is detailed in Appendix A.

### 2.9. Outcomes

Primary outcomes were safety (severe AE with a causal relationship attributable to the studied products occurring in <5% of subjects) and immunogenicity. The main criterion for immunogenicity success was a ≥4-fold anti-RBD IgG seroconversion or a ≥20% increase in RBD-ACE2 inhibitory antibodies with respect to the baseline in >55% of vaccinees in Phase 1, and >70% in Phase 2. Secondary endpoints were anti-RBD IgA titers, percentage of RBD-ACE2 inhibition and NAb against live SARS-CoV-2 variants. For IgA and NAb, seroconversion was defined as a 4-fold increase in antibody titers.

### 2.10. Procedures

Safety. This was evaluated according to a report and description of AE occurring within one-hour, post-vaccination and during the follow-up of vaccinees. Vital signs were monitored before and after vaccine administration. As vaccination was an outpatient treatment, participants were required to record any AE in an Adverse Event Diary, shared with investigators during follow-up evaluations. In Phase 1, safety monitoring was carried out for 28 days and in Phase 2 for 14 days post-vaccination. The AE type, duration, severity, outcome and causality relationship were carefully registered. Systemic AE (headache, fever, nausea and hypertension, among others) were especially sought. Local adverse reactions associated with IM (pain at the injection site, erythema and induration, among others), or with IN vaccination (as rhinitis), were also explored.

Immunogenicity. Peripheral blood samples were collected on day 0 (baseline) before the booster dose application, and 14 days after vaccination, to carry out hematology, clinical chemistry and immunogenicity assessments. Serum samples were aliquoted and stored at −20 °C until evaluation.

The immunogenicity assessment was based on seroconversion and geometric mean titers (GMT) of RBD-specific IgG and IgA antibodies, the inhibition percentage of RBD-ACE2 binding as proportions and means, and NAb levels for live SARS-CoV-2, expressed as GMT proportions.

IgG antibodies were quantified by UMELISA anti-SARS-CoV-2 RBD (Immunoassay Center, Havana, Cuba). Titers were given in arbitrary units per mL (AU/mL) with a detection limit of 1.95. ACE2-RBD antibody inhibition properties were determined using an in-house surrogate virus neutralization test inhibition (CIGB, Havana, Cuba). Results were given in inhibition percentages. The assay positivity threshold was 20%. Both methods have been described by Lemos-Pérez et al. [17].

IgA antibodies were quantified by an in-house ELISA test (CIGB, Havana, Cuba). Titers were given in AU/mL with a detection limit of 0.6 (see Appendix A).

NAb titers were detected by a viral microneutralization assay (MNA) using the live SARS-CoV-2 variants D614G (30654/21 D614G 1PVE6), Beta (34959/21 1PVE6 South Africa) and Delta (57383/21 Delta 1PVE6) in Phase 1, and D614G and Omicron (8649/22 OMICRON BA1 1PVE6) in Phase 2. Vero E6 cells (ATCC No. CRL-1586) were used for isolating passaging and MNA. Serum samples were heat-inactivated at 56 °C for 30 min and diluted in serial two-fold dilutions (from 1:10 to 1:2560) in microtiter plates. NAb titers were calculated using the Reed–Muench method, as described by Johnson et al. [18] Viral MNA was performed at the Pedro Kourí Tropical Medicine Institute (IPK, Havana, Cuba) under biosafety level 3 (BSL-3) conditions. NAb titers were calculated as the highest serum dilution without cytopathic effect on day 5 post-infection. For NAb titer <10, a final value of 5 was assigned for statistical calculations.

### 2.11. Data Management

Subject information was recorded chronologically in the medical record (MR) and the CRF. Electronic entries in CRFs using the XAVIA_SIDEC software v3.0 (UCI, Havana, Cuba) were carried out at clinical sites by clinical research co-ordinators, authorized researchers participating in the study, or by the clinical site co-ordinator. Database creation and validation were conducted at the National Clinical Trials Co-ordinating Center (CENCEC, Cuba). New entries or modifications of information collected in the CRF were recorded in auditable system traces.

### 2.12. Statistical Methods

We assessed safety and immunogenicity primary outcomes by intention-to-treat (ITT) (i.e., subjects who underwent randomization) and per-protocol (PP) analyses (i.e., subjects who, in compliance with the protocol, received a booster dose of vaccine according to protocol requirements and had serum-test results before and after immunization). To evaluate the vaccination effect, the main analysis consisted of applying a one-sided test of comparison of a proportion with a reference value. The hypotheses tested were: H0 = π ≤ π0 (null hypothesis) and H1 = π > π0 (alternative hypothesis), where π represents the proportion of subjects who met the main success criterion or vaccine responders, and π0 is the reference value for comparison. The hypotheses were examined in each of the four vaccination groups or strata conceived in both phases of the study. The level of significance (α = 0.0125), set at a quarter of the nominal value, guaranteed a global significance level α = 0.05. Pearson χ^2^ test or Fisher’s exact test were used to analyze categorical outcomes. We calculated 95% confidence intervals (95% CI) for all categorical outcomes using the Clopper–Pearson method.

Immunogenicity analyses based on the quantitative results of immune response were performed in the PP population. Anti-RBD specific IgG and IgA were determined using seroconversion rates and geometric titers (GMT). The percentage of RBD-ACE2 inhibition was determined by seroconversion rates (percentage increase ≥20%) and means. NAb titers were measured by seroconversion rates and GMT. Estimates of vaccination effect size were determined using the median of differences (MD). The 95% CIs were determined in all cases. Lilliefors or Shapiro–Wilks tests were used to prove the normality of linear model residues comparing two or more groups. The Wilcoxon matched-pairs signed rank test (*p* <0.05) was used to statistically compare pre- and post-vaccination outcomes. SAS for Windows (version 9.3), SPLUS (version 6.2) and SPSS (version 25) and GraphPad Prism (v. 9.4.1) were used for data processing and analysis.

## 3. Results

In eight months (from 15 July 2021 to 28 March 2022), 1161 subjects out of 1235 screened were included. Their distribution is shown in Figure 1. Table 1 shows participants’ demographic and baseline characteristics. No relevant imbalances were observed. Women predominated in both phases, but mainly in Phase 2 (70.5%).

Regarding control variables, subject composition in each vaccine group showed no statistically significant difference (*p* > 0.05) in terms of age and clinical category of COVID-19 disease (see Table 2).

### 3.1. Safety

In our study, AE were reported in 333 participants (51 in Phase 1 and 282 in Phase 2). Of the 747 AE reported, 126 occurred in Phase 1 and 621 in Phase 2, with causality related to vaccination of 91.3% (115/126) and 81.0% (503/621), respectively. All AE were classified as not severe, and no AE-related withdrawals occurred. Thus, the safety endpoint of <5% of participants with serious AE was met in both study phases.

Phase 1. Vaccination-related AE were reported in 42.5% (51/120) of participants, 8.3% (10/120) who had been vaccinated with Abdala, and 10% (12/120), 12.5% (15/120) and 11.7% (14/120) to groups AZ, S and D vaccinated with Mambisa, respectively. Of the 115 vaccination-related AE, 14.8% (17/115) occurred in subjects who received Abdala, 27.0% (31/115) in those who used the AZ nasal delivery device, 28.7% (33/115) in group S and 30.2% (38/126) in group D. Headaches were the most frequent AE observed in all groups, followed by low-incidence AE such as asthenia, more frequent in Abdala and Mambisa D vaccination groups. Nasal congestion and rhinitis were the most frequent local AE in the Mambisa groups (Figure 2). In terms of intensity, AE of mild intensity predominated at 73.9% (85/115), followed by moderate at 19.1% (22/115) and severe intensity at 7.0% (8/115). In this phase, 4.2% (5/120) of participants reported AE of severe intensity, including one with insomnia (Abdala group), one with hypertension, one with pruritus (Mambisa S group), one with headache, sore throat and otalgia, and one with otalgia and nasal congestion (Mambisa D group).

Phase 2. Vaccination-related AE were reported in 25.3% (263/1041) of participants, lower than in Phase 1. AE were reported in 12.4% (129/1041) of Abdala vaccinees and in 12.9% (134/1041) of Mambisa vaccinees. A total of 503 vaccine-related AE were reported, 45.5% (229/503) in Abdala vaccinees and 54.5% (274/503) in those receiving Mambisa. Regarding AE intensity, as in Phase 1, mild intensity AE predominated—77.3% (389/503) of reported AE were mild, followed by 21.3% (107/503) moderate, similar to in Phase 1. However, severe intensity AE were lower—only 1.2% (6/503) and reported by 0.4% of participants (4/1041). One Abdala vaccinee reported asthenia, myalgia and epigastric pain, and two others reported hypertension. In Mambisa groups, one participant reported headache. As in Phase 1, headaches were the predominant AE in all groups, followed by asthenia (Figure 2). As in Phase 2, in addition to the above similarities with Phase 1, pain at the injection site was a common local AE in the Abdala group, while nasal congestion and rhinitis were again the most common local AEs in the Mambisa vaccinees. Most AEs resolved spontaneously in the first 24–48 h without medication.

### 3.2. Immunogenicity

Some convalescents showed high baseline IgG titers or inhibition percentages due to a recent SARS-CoV-2 infection. Although most increased the immune response after the booster dose, when they did not comply with the seroconversion end-point levels established by protocol they were not considered as responders. Table 3 summarizes the percentages of response to vaccination, per the ITT and PP populations. All convalescents increased their titers and functionality of RBD antibodies after the booster dose with both vaccines (Figure 3).

Phase 1. In the ITT population, the percentage of Abdala vaccine responders was 83.3% (*p* = 0.0009), comparable to the results for Mambisa’s application device groups AZ and S. Group D showed 73.3% (*p* = 0.0218). Similar results were obtained in the PP population. All responder percentages were significant—>55% the defined endpoint (Table 3).

Table 4 summarizes the immunogenicity results in both study phases.

Serum anti-RBD IgG titers increased significantly in all vaccination groups (Figure 3A). In subjects receiving Abdala, the vaccine effect given by the MD in the IgG titers was 463.8 AU/mL (95% CI 173.2–1473). The Mambisa application device group AZ showed an MD of 22.0 AU/mL (95% CI 11.4–122.3); group S showed the highest MD, 57.5 AU/mL (95% CI 28.6–186.3), followed by group D, 45.5 AU/mL (95% CI 12.7–89.89) (Table 4).

Serum anti-RBD IgA titers increased in all vaccination groups (Figure 3B). However, when comparing Mambisa group devices, group D showed the lowest MD and seroconversion rates, likely due to the delivery method.

Antibody functionality given by inhibition percentages also increased after booster doses for both vaccines (Figure 3C). Abdala vaccinees showed the highest increase with an MD of 70.3% (95% CI 59.0–84.9). The Mambisa group AZ showed an MD of 53% (95% CI 30.7–65.6), similar to group S, but group D showed the lowest (Table 4).

NAb titers against SARS-CoV-2 variants significantly increased in all vaccination groups (Figure 3D–F panels), but mainly against the Delta variant in all vaccine groups. For both vaccines, the lowest response was observed against the D614G strain, and differences between the responses against variants, from highest to lowest, were as follows: Delta > Beta > D614G (Table 4). Although results for the three nasal devices did not differ greatly, AZ was chosen for Phase 2 due to its easy availability.

Phase 2. For the ITT population, the percentage of responders was 85.6% (*p* < 0.0001) for Abdala and 71.2% (*p* = 0.2753) for Mambisa. These were similar for the PP population. All percentages were higher than 70%, the established success criterion (Table 3).

As a whole, there were a similar proportion of responders in both age groups: 81.9% for those aged ≤60 years (*p* < 0.0001) and 80.8% for >60 years (*p* < 0.0001). All responder percentages were higher than 70% (Table 3).

Anti-RBD IgG titers increased significantly after the booster dose in both vaccination groups (Figure 3G). In both of them, MD values were higher than those observed in Phase 1 (Table 4).

Anti-RBD IgA titers also increased after the booster dose (Figure 3H). The Abdala group showed an MD of 3416 AU/mL (95% CI 3080–3803), and the Mambisa group 2143 AU/mL (95% CI 1658–2485).

Inhibition of RBD-ACE2 binding increased significantly after booster application (Figure 3I). The Abdala group showed an MD in inhibitory percentages of 59.0% (95% CI 53.0–63.4); the Mambisa group showed 34.5% (95% CI 29.3–38.1).

NAb titers were high after the booster dose (Figure 3J,K panels) for both vaccination groups. NAb titers against the D614G variant almost doubled those against Omicron (Table 4).

When immune response was compared by age group, a similar statistically significant increase in anti-RBD IgG titers was observed for both vaccines. However, anti-RBD IgG and IgA GMT were slightly higher in older vaccinees on day 14 (Figure 4).

Regarding anti-RBD IgA titers, Abdala vaccinees >60 years had an MD of 3680 AU/mL (95% CI 3146–4167), similar to those ≤60 years. However, Mambisa vaccinees >60 years showed an MD of 2460 (95% CI 1811–3133), nearly double that of those ≤60 (Table 4).

Results of the RBD-ACE2 binding inhibition were similar in both age groups with both vaccines. Abdala vaccinees’ seroconversion rates showed differences, probably due to higher baseline inhibition levels in individuals >60 years. After vaccination, inhibition percentages were similar in both age groups.

Regarding the influence of age on NAb results, baseline neutralizing titers were low against D614G and Omicron. After the booster dose, a significant NAb increase was observed due to vaccination. This increase was higher in subjects >60 years, both against D614G and Omicron variants and with both vaccines (Figure 4D,E,I,J panels).

NAb against D614G increased both in Abdala and Mambisa vaccinees, showing no significant differences in individuals ≤60 and >60 years. Individuals >60 years responded as well as those ≤60 years in both immunization groups. In the Abdala group, the neutralization titers were slightly higher in those >60 years (Table 4).

NAb titers against Omicron in Abdala vaccinees >60 years doubled the result of subjects ≤60 after vaccination. However, Mambisa age groups showed very similar MD values, the highest being 150.0 (95% CI 20.0–290.0), corresponding to those >60 years.

In summary, the expected seroconversion outcome was observed in all immunogenicity variables of both vaccines (Table 4).

## 4. Discussion

Although most of the world’s population is recovering from the COVID-19 pandemic, having a vaccine to boost immunity against SARS-CoV-2 variants is of the utmost importance.

In our study, a booster dose was applied to COVID-19 convalescents to evaluate the capacity of two vaccines to enhance immune response to SARS-CoV-2. As these products had different formulations and administration routes, their mechanisms for inducing response at the immune system’s local and systemic levels differed too. Thus, the results obtained should not be expected to be exactly the same, but should meet the study’s established success outcomes.

Regarding safety, no serious AE were recorded. AE reports were minimal, mostly of mild intensity and short duration, which resolved spontaneously. Headaches were a common systemic vaccine AE that were observed for both vaccines. Pain at the injection site was the most frequent local reaction for Abdala, and rhinitis was for Mambisa, as expected for IM and IN vaccines, but of very low incidence. These safety results were consistent with previous studies conducted with both vaccines [8,9,12].

In previous Phases 1–2 and Phase 3 clinical trials in seronegative adult populations, Abdala was demonstrated to be safe and well tolerated in its three-dose schedule, and vaccine reactogenicity was dose-independent [8,12]. Pain at the injection site is consistent with findings for other COVID-19 IM vaccines [19]. Abdala’s safety has also been evaluated in children and adolescents aged 3–18 [14], in pregnant women [15] and as a booster dose for seronegative individuals [8], with the same results.

The emergence of new SARS-CoV-2 variants, such as Omicron and its sub-variants, has revealed substantial immune evasion in serum samples from infected persons [20], or in seronegative persons who received two-dose vaccine schedules [21]. A booster dose can substantially increase neutralizing antibodies and surrogate markers, such as anti-RBD titers. It also allows an improved immune response to different SARS-CoV-2 variants, and can reduce reinfection rates or, if reinfection does occur, at least ensure milder symptoms. The booster dose also helps to stimulate immune response among people who do not respond to standard vaccination schedules, such as immunocompromised individuals [22].

The proposed success criteria for vaccine immunogenicity were met in both phases of this clinical trial. A total of 80% of those vaccinated with Abdala and 75% receiving Mambisa showed an increase in anti-RBD IgG titers and/or at least RBD-ACE2 inhibition percentages.

In previous studies, Abdala was shown to be highly immunogenic [8,10,23]. One of the reasons proposed is the presence of mannose structures in the recombinant RBD protein expressed in the yeast. These carbohydrate structures act as strong adjuvants and are easily detected by the receptors of antigen-presenting cells, promoting antigen presentation and T cell activation [8,24].

The humoral response induced by the IM route will protect the lower respiratory tract, but will not activate tissue-specific resident memory cells, necessary for long lasting protection in the upper respiratory tract and reduction of transmission rates [25,26]. Mucosal immunization can induce extensive adaptive immune responses, characterized by secretory IgA antibodies [27]. It can also effectively stimulate a potent systemic immune response and generate serum antibodies with neutralizing properties, reflecting the interaction between the mucosal and systemic immune systems [27].

We observed that COVID-19 convalescents boosted with Abdala demonstrated a strong systemic humoral response. In both study phases, anti-RBD IgG GMT increased more than 40-fold after the booster dose, and in the case of the mean of inhibition percentages, this increased by more than 50%. Anti-RBD IgA titers also increased.

Since the three devices used for nasal delivery in the exploratory study (Phase 1) exhibited similar results, the most readily available device was chosen for Phase 2. This atomizer permitted precision dosing of the vaccine into the atrium or anterior region of the nasal respiratory epithelium, allowing immunogen contact in the nasal environment [28].

Although the study’s endpoints were based exclusively on systemic immunogenicity variables, Mambisa vaccinees showed a strong humoral immune response after the booster dose. In Phase 2, anti-RBD IgG titers increased more than five-fold, and in the case of antibody inhibition capacity, they increased by more than 30%. Also, anti-RBD IgA titers increased by more than 2000 AU/mL.

Mambisa’s HBcAg immunopotentiating capacity favored a local and systemic immune response to the vaccine through multiple toll-like receptor signaling pathways [29]. Similar results of increased systemic humoral response from IN vaccination have been observed elsewhere, including with vaccines based on adenovirus vectors, such as the ChAdOx1 nCoV-19 vaccine [30]. For example, in a phase 3 study, the BBV154 nasal vaccine showed higher humoral immune response compared to Covaxin (BBV152), a vaccine comprised of inactivated SARS-CoV-2 and administered IM in two doses. At 42 days post-vaccination, participants showed 2- and 1.5-fold increases in serum IgG and IgA antibody levels, respectively [31].

Regarding viral neutralization, both vaccines showed significantly increased NAb titers after the booster dose. In Phase 1, different responses to the different variants were observed, with the highest NAb titers being against Delta and Beta variants, and the lower titers against D614G. This was interesting because, generally, COVID-19 vaccines have shown a lower neutralizing response to the Beta variant [32]. In this study, the booster dose was found to significantly enhance responses to various variants, with particularly substantial increases observed against the Beta and Delta variants. An explanation could be that Beta and Delta variants were the ones that had circulated the most in Cuba before the recruitment of volunteers for this study, and the results show the ability of the vaccines to recall B cells that produce high-affinity antibodies against the epitopes conserved between the different variants. These findings underscore the critical role of booster doses for convalescent individuals, and highlight the robust response generated by hybrid immunity. Additionally, the significant increase in response to Delta can be explained by its local circulation—it was dominant at the time when most of the volunteers were infected with the virus [33].

In Phase 2, post-booster antibodies showed neutralization activity against Omicron in both vaccination groups. Noteworthy is the fact that convalescents enrolled in the study were infected before November 2021, when Omicron variants were not yet circulating in Cuba [32]. This corroborates previous reports referencing Omicron’s immune evasion being less pronounced in individuals with hybrid immunity as a result of prior infection and vaccination [5].

We observed no difference in compliance with the trial success criteria for subjects ≤60 and >60 years of age in either vaccine group. As analyzed in Phase 2, systemic anti-RBD IgG and IgA levels were higher in those aged >60 years. RBD-ACE2 inhibitory activity percentages were similar for both age groups. However, older adults also showed higher NAb titers against D614G and Omicron variants, with titers against the ancestral variant being two-fold higher. In those vaccinated with Abdala, MD values were twice as high in older adults compared to younger individuals.

The effectiveness of COVID-19 vaccines depends largely on the titers of SARS-CoV-2-neutralizing antibodies [34], regardless of vaccine type, in conjunction with a booster dose effective at increasing these antibody titers in older adults [35]. In our study, the best neutralizing antibody response was seen in convalescents >60 years of age, in both IN and IM routes. It is well known that older adults are most susceptible to infections, due to more restricted antibody and T cell repertoires, which limits de novo generation of antibodies [36]. However, in previous studies with the BNT162b2 vaccine, enhancement of a strong memory B cell response was observed in individuals >60 years of age after booster doses, showing a greater increase than in younger subjects [37]. A study in 2022 postulated that the presence of an increased population of RBD-specific memory (CD27^+^ CD21^–^) B cells in older adults may indicate prolonged persistence of the virus and, therefore, B-cell activation following SARS-CoV-2 infection [38]. This could explain the higher humoral response to vaccination in this age group—results that were unrelated to convalescents’ records of disease severity. The best neutralization results observed in subjects >60 years of age in our study corroborate results obtained when comparing the levels of IgA and IgG antibodies, with anti-RBD activity due to the booster dose being the most responsible for the neutralizing capacity of the samples.

Our study has two main limitations. First, we were unable to measure secretory IgA and Tissue-resident memory T cells or TRM cells at the mucosal level, a helpful analytical tool to demonstrate upper respiratory tract immune status before and after the booster dose. Second, anti-RBD IgG titers were expressed in AU/mL since, at the time of the evaluations, the reference material employed by the UMELISA anti-SARS-CoV-2 RBD was not yet calibrated against the WHO International Standard for anti-SARS-CoV-2 immunoglobulin.

This is the first clinical-study report featuring a nasal subunit vaccine targeting COVID-19. Additionally, this study also demonstrated the capacity of the particulate antigen from the hepatitis B virus nucleocapsid to stimulate the immune response when administered through the nasal route.

Abdala and Mambisa proved to be highly immunogenic. A booster dose in individuals previously infected with the SARS-CoV-2 strengthened the immune response, resulting in better protection against new viral variants. Additionally, both vaccines are easy to apply and store, since they do not need low storage temperatures required for some other COVID-19 vaccines.

## 5. Conclusions

Administration of the Abdala IM vaccine and Mambisa IN vaccine as booster doses for COVID-19 convalescents was safe and well tolerated. Both products surpassed the clinical trial’s immunogenicity endpoints. The booster dose with both vaccines strengthened participants’ immune responses, including among older adults.

## Figures and Tables

**Figure 1 vaccines-12-01001-f001:**
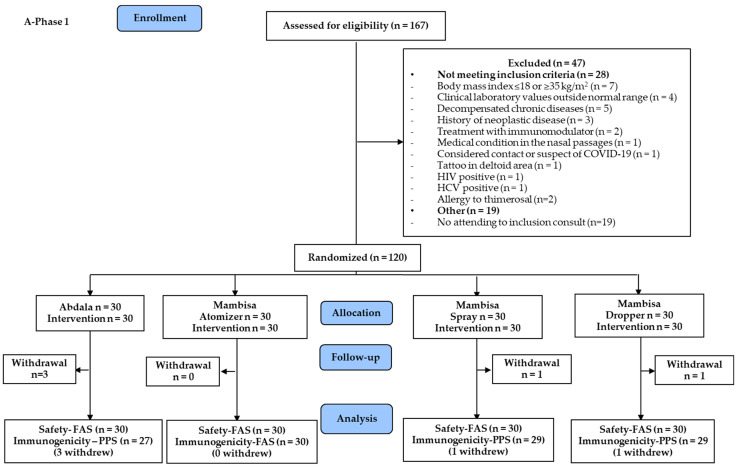
Study flow diagram. ** Other exclusion criteria were: not attending the inclusion consultation (19 subjects in (**A**) Phase 1), immunodeficiency and unconfirmed COVID-19 disease by PCR (1 subject each in (**B**) Phase 2). FAS, full analysis set; PPS, per-protocol set. In Phase 2, the intervention was discontinued in one participant due to uncontrolled hypertension. All withdrawals were due to voluntary abandonment.

**Figure 2 vaccines-12-01001-f002:**
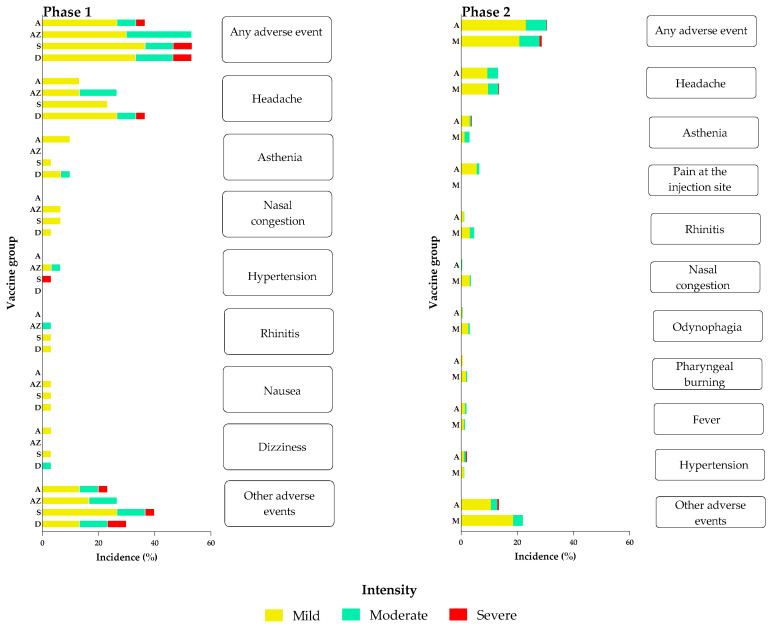
Percentage of participants in each vaccination group according to the most frequently detected adverse events with certain or probable causality due to vaccination. A: Abdala vaccine candidate. M: Mambisa vaccine candidate. Mambisa nasal vaccine delivery devices: AZ: intranasal atomization device. D: dropper. S: nasal spray. Participants reporting 0 adverse events make up the remainder of the 100% in each vaccination group. Bar colors represent adverse events’ intensity levels. Other adverse events: Phase 1: odynophagia, fever, diarrhea, pruritus, otalgia and nasal itching, among others. Phase 2: myalgia, drowsiness, pharyngeal burning, cough with expectoration and rhinorrhea, among others.

**Figure 3 vaccines-12-01001-f003:**
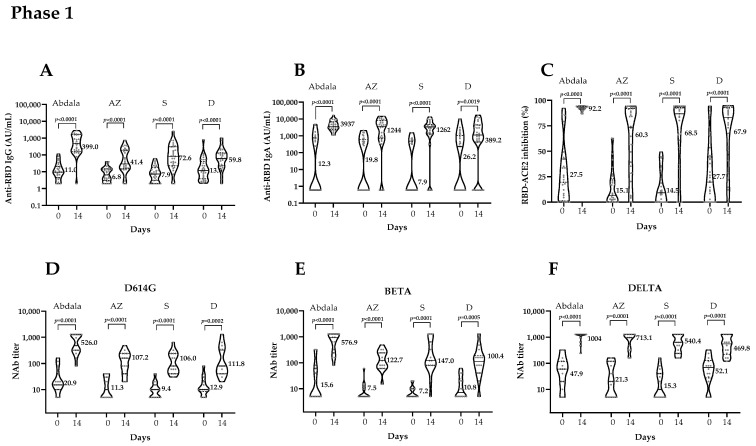
Quantitative results for immunogenicity variables in both study phases, at the beginning (Day 0) and 14 days after the booster dose (Day 14). ACE2: Angiotensin-converting enzyme 2. AU/mL: arbitrary units per mL. AZ, S and D: Mambisa nasal application devices (atomizer, spray and dropper). NAb: neutralizing antibody titers. RBD: receptor binding domain. Brackets contain the results of the Wilcoxon matched-pairs signed rank test. Numbers represent GMT (mean in the case of inhibition). Phase 1: graphs represent vaccination groups: Abdala (n = 27) and Mambisa AZ (n = 30), S (n = 29) and D (n = 29). Panel (**A**): anti-RBD IgG titers. Panel (**B**): anti-RBD IgA titers. Panel (**C**): inhibition percentages of RBD-ACE2 binding. Panels (**D**–**F**): NAb titers in Abdala (n = 15) and Mambisa AZ (n = 15), S (n = 15) and D (n = 14) vaccination groups against D614G, Beta and Delta SARS-CoV-2 variants, respectively. Phase 2: graphs represent vaccination groups: Abdala (n = 509) and Mambisa (n = 494). Panel (**G**): anti-RBD IgG titers. Panel (**H**): anti-RBD IgA titers. Panel (**I**): inhibition percentages of RBD-ACE2 binding. Panels (**J**,**K**): NAb titers for Abdala (n = 50) and Mambisa (n = 52) vaccination groups against D614G and Omicron (BA.5) SARS-CoV-2 variants, respectively.

**Figure 4 vaccines-12-01001-f004:**
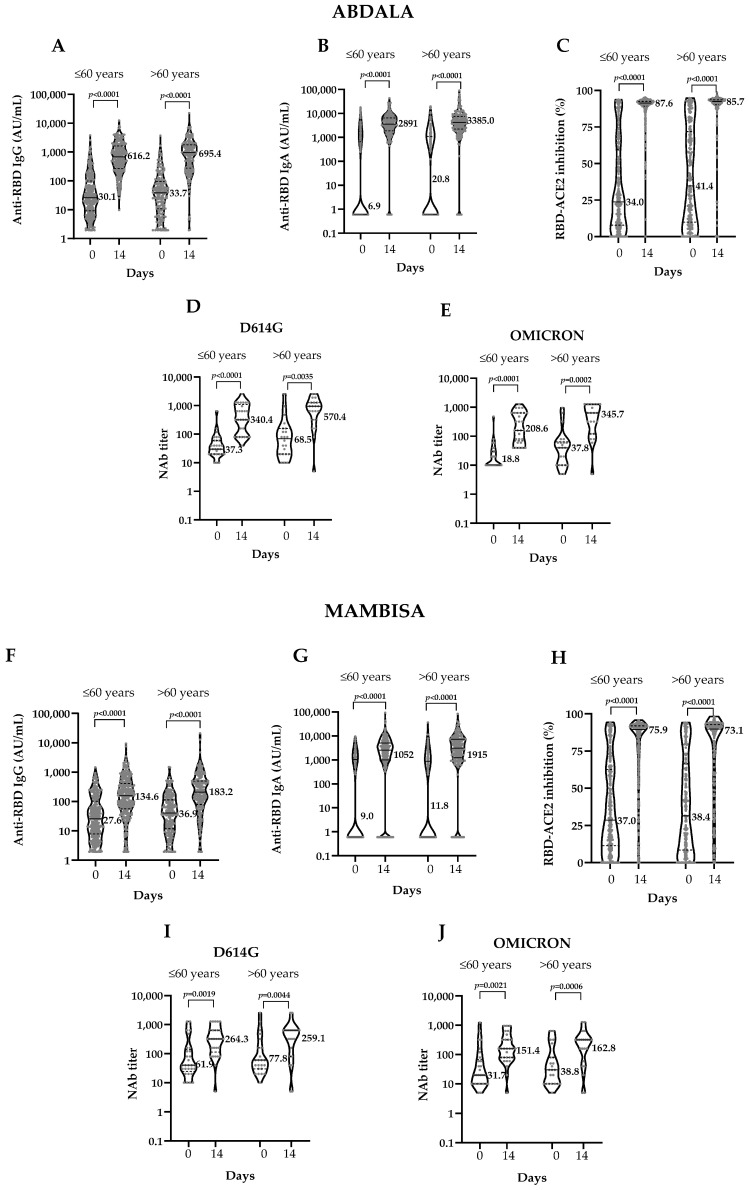
Phase 2 immunogenicity results by age group (≤60 years and >60 years) at the beginning (Day 0) and 14 days after the booster dose (Day 14). ACE2: Angiotensin-converting enzyme 2. AU/mL: arbitrary units per mL. NAb: neutralizing antibody titers. RBD: receptor binding domain. Graphs represent vaccination groups. Abdala (Panels (**A**–**E**)) and Mambisa (Panels (**F**–**J**)). Panels (**A**,**I**,**B**,**G**) show RBD IgG and IgA antibody titers, respectively. Panels (**C**,**H)** show inhibition percentages of RBD-ACE2 binding. Panels (**D**,**E**,**I**,**J**) show NAb titers against D614G (panels (**D**,**I**)) and Omicron (Panels (**E**,**J**)) SARS-CoV-2 variants, respectively. Brackets contain the results of the Wilcoxon matched-pairs signed rank test. Numbers represent GMT (mean in the case of inhibition).

**Table 1 vaccines-12-01001-t001:** Participant demographics and COVID-19 disease characteristics.

Phase 1							
		Group					
Variable		Abdala	Mambisa (Three Nasal Application Devices)				
			Atomizer	Spray	Dropper	Subtotal	Total
N		30	30	30	30	90 (100)	120 (100)
Sex—n (%)	Female	18 (60.0)	14 (46.7)	15 (50.0)	23 (76.7)	52 (57.8)	70 (58.3)
	Male	12 (40.0)	16 (53.3)	15 (50.0)	7 (23.3)	38 (42.2)	50 (41.7)
Age	Years	47.9 ± 15.1	50.9 ± 13.1	49.5 ± 17.1	48.6 ± 15.7	49.1 ± 14.5	49.2 ± 15.2
Skin color—n (%)	White	19 (63.3)	18 (60.0)	17 (56.7)	20 (66.7)	55 (61.1)	74 (61.7)
	Mulatto	3 (10.0)	8 (26.7)	4 (13.3)	4 (13.3)	16 (17.8)	19 (15.8)
	Black	8 (26.67)	4 (13.3)	9 (30.0)	6 (20.0)	19 (21.1)	27 (22.5)
BMI ± SD	kg/m^2^	26.5 ± 4.0	27.4 ± 3.7	26.6 ± 3.0	27.2 ± 4.0	26.7 ± 3.6	26.9 ± 3.7
Age group—n (%)	≤60 years	24 (80.0)	25 (83.3)	23 (76.7)	23 (76.7)	71 (78.9)	95 (79.2)
	>60 years	6 (20.0)	5 (16.7)	7 (23.3)	7 (23.3)	19 (21.1)	25 (20.8)
*p*-value		0.9065					
COVID-19 symptoms—n (%)	Asymptomatic	9 (30.0)	8 (26.7)	10 (33.3)	7 (23.3)	25 (27.8)	34 (28.3)
	Symptomatic	21 (70.0)	22 (73.3)	20 (13.3)	23 (76.7)	65 (72.2)	86 (71.7)
*p*-value		0.8445					
COVID-19 disease severity—n (%)	Severe	2 (9.5) *	1 (4.5) *	2 (10.0) *	5 (21.7) *	8 (8.9) *	10 (11.6) *
	Mild	12 (57.4) *	19 (86.4) *	11 (55.0) *	12 (52.2) *	42 (46.7) *	54 (62.8) *
	Moderate	7 (33.3) *	2 (9.1) *	7 (35.0) *	6 (26.1) *	25 (27.8) *	22 (25.6) *
*p*-value		0.1582					
Phase 2							
		Group					
Variable		Abdala		Mambisa		Total	
N		527		514		1041 (100)	
Sex—n (%)	Female	370 (70.2)		364 (70.8)		734 (70.5)	
	Male	157 (29.8)		150 (29.2)		307 (29.5)	
Age	Years	55.3 ± 13.7		55.0 ± 14.1		55.1 ± 13.9	
Skin color—n (%)	White	397 (75.3)		384 (74.7)		781 (75.0)	
	Mulatto	49 (10.0)		56 (26.7)		105 (10.1)	
	Black	81 (26.67)		74 (13.3)		155 (14.9)	
BMI ± SD	kg/m^2^	26.9 ± 4.2		26.6 ± 4.1		26.7 ± 4.1	
Age group—n (%)	≤60 years	310 (58.8)		308 (59.9)		618 (59.4)	
	>60 years	217 (41.2)		206 (40.1)		423 (40.6)	
*p*-value		0.3830					
COVID-19 symptoms—n (%)	Asymptomatic	27 (5.1)		35 (6.8)		62 (6.0)	
	Symptomatic	500 (94.9)		479 (93.2)		979 (94.0)	
*p*-value		0.1543					
COVID-19 disease severity—n (%)	Severe	38 (7.6) *		50 (10.4) *		88 (9.0) *	
	Mild	305 (61.0) *		276 (57.6) *		581 (59.3) *	
	Moderate	157 (31.4) *		153 (31.9) *		310 (31.7) *	
*p*-value		0.2608					

BMI: Body-mass index (weight in kilograms divided by the square of the height in meters; calculation was based on the weight and height measured at the time of screening). N: number of participants in the specified group or the total sample (also the denominator for percentage calculations). n: number of participants with the specified characteristic. SD: standard deviation. * Percentage calculations based on symptomatic individuals. *p*-values are for comparisons between vaccination groups.

**Table 2 vaccines-12-01001-t002:** COVID-19 disease category stratification in participants according to age groups in PP population.

Phase 1									
		Group							
Variable		Abdala		Mambisa (Three Nasal Application Devices)					
				Atomizer		Spray		Dropper	
Age		≤60 years	>60 years	≤60 years	>60 years	≤60 years	>60 years	≤60 years	>60 years
N		22	5	25	5	22	7	22	7
Asymptomatic—n (%)		8 (36.4)	0 (0.0)	8 (32.0)	0 (0.0)	7 (31.8)	3 (42.8)	7 (31.8)	0 (0.0)
Symptomatic—n (%)		14 (63.6)	5 (100.0)	17 (68.0)	5 (100.0)	15 (57.1)	4 (57.1)	15 (68.2)	7 (100.0)
*p*-value		0.2798		0.2868		0.6647		0.1470	
COVID-19 disease severity—n (%)	Severe	1 (7.1) *	1 (20.0) *	1 (5.9) *	0 (0.0) *	1 (6.7) *	1 (25.0) *	2 (13.3) *	3 (42.8) *
	Mild	10 (71.4) *	1 (20.0) *	15 (88.2) *	4 (80.0) *	10 (66.7) *	1 (25.0) *	9 (60.0) *	2 (28.6) *
	Moderate	3 (21.4) *	3 (60.0) *	1 (5.9) *	1 (20.0) *	4 (26.7) *	2 (50.0) *	4 (26.7) *	2 (28.6) *
*p*-value		0.4213		>0.9999		0.3860		0.2743	
Phase 2									
Variable		Group							
		Abdala				Mambisa			
Age		≤60 years		>60 years		≤60 years		>60 years	
N		298		211		294		200	
Asymptomatic—n (%)		19 (6.4)		7 (3.3)		21 (7.1)		12 (6.0)	
Symptomatic—n (%)		279 (93.6)		204 (96.6)		273 (92.8)		188 (94.0)	
*p*-value		0.1534				0.7149			
COVID-19 disease severity—n (%)	Severe	22 (7.9) *		15 (7.3) *		25 (9.1) *		21 (11.2) *	
	Mild	171 (61.3) *		123 (60.3) *		170 (62.3) *		97 (51.6) *	
	Moderate	86 (30.8) *		66 (32.3) *		78 (28.6) *		70 (37.2) *	
*p*-value		0.9276				0.0726			

N: number of participants in the specified group or the total sample (this value is the denominator for percentage calculations). n: number of participants with the specified characteristic. * Percentage calculations based on symptomatic individuals. *p*-values are for comparisons between vaccination groups.

**Table 3 vaccines-12-01001-t003:** Percentages of responders to vaccination in each group in ITT and PP populations *.

Phase 1					
	Group				
Variable	Abdala	Mambisa (Three Nasal Delivery Devices)			Total
		Atomizer (AZ)	Spray (S)	Dropper (D)	
	ITT population				
Responders-n/N (%; 95% CI)	25/30 (83.3; 65.2–94.3)	25/30 (83.3; 65.2–94.3)	25/30 (83.3; 65.2–94.3)	22/30 (73.3; 54.1–87.7)	97/120 (80.8; 72.6–87.4)
*p*-value	0.0009	0.0009	0.0009	0.0218	<0.0001
	PP population				
Responders-n/N (%; 95% CI)	25/27 (92.6; 75.7–99.1)	25/30 (83.3; 65.2–94.3)	25/29 (86.2; 68.3–96.1)	22/29 (80.0; 61.0–92.4)	97/115 (84.3; 76.4–90.4)
*p*-value	0.0000	0.0009	0.0007	0.0119	<0.0001
Phase 2					
	Group				
Variable	Abdala		Mambisa		Total
	ITT population				
Responders-n/N (%; 95% CI)	451/527 (85.6; 82.3–88.5)		366/514 (71.2; 67.1–75.1)		817/1041 (78.5; 75.7–81.0)
*p*-value	<0.0001		0.2753		<0.0001
	PP population				
Responders-n/N (%; 95% CI)	451/509 (88.6; 85.5–91.2)		366/494 (74.1; 70.0–77.9)		817/1003 (81.4; 78.9–83.8)
*p*-value	<0.0001		0.0237		<0.0001
Age	≤60 years				
Responders-n/N (%; 95% CI)	263/298 (88.2; 84.0–91.6)		222/294 (75.5; 70.2–80.3)		485/592 (81.9; 78.6–84.9)
*p*-value	<0.0001		0.0196		<0.0001
Age	>60 years				
Responders-n/N (%; 95% CI)	188/211 (89.1; 84.1–93.0)		144/200 (72.0; 65.2–78.1)		332/411 (80.8; 76.6–84.5)
*p*-value	<0.0001		0.2685		<0.0001

ITT: Intention to treat. N: number of participants in the specified group or the total sample. n: number of participants with the specified characteristic. PP: per protocol. * Immunogenicity endpoints for responders were ≥4-fold anti-RBD IgG seroconversion, or ≥20% of inhibitory antibodies (inhibition of the RBD-ACE2 binding) from baseline in >55% of vaccinees in Phase 1 and >70% in Phase 2. *p*-values are for one-sided test of comparison of a proportion (α = 0.0125) with the defined endpoints mentioned above for each study phase.

**Table 4 vaccines-12-01001-t004:** Overall immunogenicity results by vaccination group in both study phases.

Phase 1				
	Group			
Variable	Abdala	Mambisa (Three Nasal Application Devices)		
		Atomizer	Spray	Dropper
	Anti-RBD IgG			
Seroconversion rate-n/N (%; 95% CI)	25/27 (92.6; 75.7–99.1)	21/30 (70; 50.6–85.3)	20/29 (68.9; 49.1–84.7)	15/29 (51.7; 32.5–70.5)
*p*-value		0.2646		
MD (95% CI)	463.8 (173.2–1473)	22.0 (11.4–122.3)	57.5 (28.6–186.3)	45.5 (12.7–89.89)
	Anti-RBD IgA			
Seroconversion rate-n/N (%; 95% CI)	23/27 (85.2; 66.3–95.8)	22/30 (73.3; 54.1–87.7)	24/29 (82.7; 64.2–94.1)	14/29 (48.3;29.5–67.9)
*p*-value		0.0142		
MD (95% CI)	3333 (2357–6104)	2635 (813.9–5947)	2825 (1345–4245)	650.4 (0.0–2929)
	Inhibition of the RBD-ACE2 binding			
Seroconversion rate-N/n (%, 95% CI)	23/27 (85.2; 66.3–95.8)	23/30 (84.1; 66.2–94.8)	24/29 (82.7; 64.2–94.1)	18/29 (62.1; 42.3–79.3)
*p*-value		0.1826		
MD (95% CI)	70.3 (59.0–84.9)	53.0 (30.7–65.6)	57.7 (40.1–75.5)	40.8 (4.1–73.9)
	Neutralizing antibodies to live SARS-CoV-2 variants			
Variant	D614G			
Seroconversion rate-n/N (%; 95% CI)	14/15 (93.3; 68.1–99.8)	14/15 (93.3; 68.1–99.8)	14/15 (93.3; 68.1–99.8)	12/14 (85.7; 57.2–98.2)
*p*-value		0.7152		
MD (95% CI)	475 (305–1240)	155 (30–220)	75 (35–230)	57.5 (20–475)
Variant	Beta			
Seroconversion rate-n/N (%; 95% CI)	13/15 (86.7; 59.6–98.3)	14/15 (93.3; 68.1–99.8)	14/15 (93.3; 68.1–99.8)	11/14 (78.5;49.1–95.3)
*p*-value		0.3561		
MD (95% CI)	955 (235–1260)	115 (55–235)	115 (70–620)	87.5 (15–280)
Variant	Delta			
Seroconversion rate-n/N (%; 95% CI)	15/15 (100; 78.2–100)	13/15 (86.7; 59.6–98.3)	15/15 (100; 78.2–100)	12/14 (85.7; 57.2–98.2)
*p*-value		0.3201		
MD (95% CI)	1160 (800–1260)	955 (315–1260)	560 (235–1220)	360 (155–1160)
Phase 2				
	Group			
Variable	Abdala		Mambisa	
	Anti-RBD IgG			
Seroconversion rate-N/n (%; 95% CI)	408/509 (80.1; 76.4–83.5)		261/494 (52.1; 48.3–57.3)	
MD (95% CI)	667.0 (519.4–804.0)		95.8 (78.0–125.0)	
	Anti-RBD IgA			
Seroconversion rate-N/n (%; 95% CI)	387/509 (76.0; 72.1–79.7)		326/494 (66.0; 61.8–70.4)	
MD (95% CI)	3416 (3080–3803)		2143 (1658–2485)	
	Inhibition of the RBD-ACE2 binding			
Seroconversion rate-N/n (%; 95% CI)	381/509 (74.8; 70.8–78.5)		317/494 (64.2; 48.5–57.5)	
MD (95% CI)	59.0 (53.0–63.4)		34.5 (29.3–38.1)	
	Neutralizing antibodies to live SARS-CoV-2 variants			
Variant	D614G			
Seroconversion rate-N/n (%; 95% CI)	43/50 (86.0; 64.0–88.5)		30/52 (57.7; 41.3–69.5)	
MD (95% CI)	480.0 (280.0–630.0)		220.0 (50.0–310.0)	
Variant	Omicron (BA.5)			
Seroconversion rate-N/n (%; 95% CI)	39/50 (78.0; 64.0–88.5)		34/52 (65.4; 48.9–76.3)	
MD (95% CI)	280.0 (140.0–580.0)		150.0 (60.0–240.0)	
Phase 2. Immunogenicity Results According to Age Groups				
	Group			
Variable	Abdala		Mambisa	
Age	≤60 years	>60 years	≤60 years	>60 years
	Anti-RBD IgG			
Seroconversion rate-N/n (%; 95% CI)	223/298 (74.8, 69.5–79.7)	173/211 (81.9; 73.1–86.9)	148/294 (50.3; 44.4–56.2)	101/200 (50.5; 43.4–57.6)
*p*-value	0.0656		>0.9999	
MD (95% CI)	541.0 (445.9–720.5)	845.0 (645.4–966.2)	79.3 (58.6–106.9)	125.7 (89.8–195.0)
	Anti-RBD IgA			
Seroconversion rate-N/n (%; 95% CI)	234/298 (78.5; 73.3–83.0)	153/211 (72.5; 66.0–78.4)	189/294 (64.3; 58.5–69.7)	137/200 (68.5; 61.4–74.9)
*p*-value	0.1399		0.3840	
MD (95% CI)	3212 (2712–3604)	3680 (3146–4167)	1893 (1435–2376)	2460 (1811–3133)
	Inhibition of the RBD-ACE2 binding			
Seroconversion rate-N/n (%; 95% CI)	236/298 (79.2; 74.1–83.7)	145/211 (68.7; 62.0–74.9)	197/294 (67.0; 60.6–71.7)	120/200 (60.0; 52.8–66.8)
*p*-value	0.0094		0.1262	
MD (95% CI)	63.5 (58.9–66.7)	44.0 (31.3–58.2)	35.4 (29.6–45.5)	31.8 (24.1–38.1)
	Neutralizing antibodies to live SARS-CoV-2 variants			
Variant	D614G			
Seroconversion rate-N/n (%; 95% CI)	27/29 (93.1; 77.2–99.1)	16/21 (76.2; 52.8–91.8)	24/29 (82.7; 64.2–94.1)	16/23 (69.6; 47.1–86.8)
*p*-value	0.1153		0.3290	
MD (95% CI)	294.0 (120.0–620.0)	610.0 (40.0–1240)	200.0 (40.0–310.0)	290.0 (0.0–560.0)
Variant	Omicron (BA.5)			
Seroconversion rate-N/n (%; 95% CI)	25/29 (86.2; 68.3–96.1)	14/21 (66.7; 43.1–85.4)	20/29 (69.0; 49.2–84.7)	14/23 (60.9; 38.6–80.3)
*p*-value	0.1658		0.5711	
MD (95% CI)	150.0 (60.0–600.0)	320.0 (80.0–950.0)	120.0 (50.0–260.0)	150.0 (20.0–290.0)

ACE2: angiotensin-converting enzyme 2. MD: median of differences. RBD: receptor binding domain. Seroconversion was considered ≥4-fold increase in anti-RBD IgG, anti-RBD IgA, or NAb titer, or ≥20% of inhibitory antibodies (inhibition of the RBD-ACE2 binding). *p*-value is for seroconversion rate comparison between Mambisa vaccination groups in Phase 1 and age strata in each vaccination group in Phase 2.

## Data Availability

The protocol and study data are available upon request directed to the corresponding authors (gilda.lemos@cigb.edu.cu and gerardo.guillen@cigb.edu.cu). Requested data can be shared through a secure online platform after signing a data access agreement.

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
