# Peer review of "Safety and Immunogenicity of the Intranasal Vaccine Candidate Mambisa and the Intramuscular Vaccine Abdala Used as Booster Doses for COVID-19 Convalescents: A Randomized Phase 1–2 Clinical Trial"

_vaccines, 2024, doi:10.3390/vaccines12091001_

Round 1
Reviewer 1 Report
Comments and Suggestions for Authors
The study is a clinical trial to evaluate the safety and immunogenicity of booster doses of Mambisa and Abdala vaccines in people who have recovered from COVID-19. Based on the available data, both vaccines are safe and effective. Although vaccines based on prototype virus strain is far behind the situation and could not be used against epidemic now, but there are still some interesting finding,but still I have some concerns, details are as following,
Major:
1. Why Omicron-BA.5 was chosen instead of other variants such as XBB, EG.5, JN.1 as to explore the level of neutralizing antibodies after virus in epidemic.
2. Although the immune response of COVID-19 convalescents increased 14 days after vaccination with both vaccines, it seems that the neutralizing antibody titer against Omicron-BA.5 did not increase much.
3. The number of samples tested for neutralizing antibodies is far less than that for other test indicators.
4. What is cellular response look like in the trial? Have the authors ever tested?
5. Inadequate detail is provided in the population - Have you ever considered disease classfication under infection status or vaccination history before enrolled in this study.
Minor:
The table does not strictly correspond to the position the author wants to describe, making it difficult for readers to understand. For example, the positions of Day 0 and Day 14 in Table 4 are confusing.
Comments on the Quality of English Language
English is fine.
Author Response
Thank you very much for reviewing this manuscript
Comments 1: Why Omicron-BA.5 was chosen instead of other variants such as XBB, EG.5, JN.1 as to explore the level of neutralizing antibodies after virus in epidemic.
Response 1: The clinical trial began in July 2021, and Omicron VOC was first detected in the country by the end of 2021. Therefore, the only available strains for neutralization experiments in Phase I were Alpha, Beta and Delta strains. During Phase II, the only Omicron subvariant available for neutralization experiments was BA.5. The variants XBB, EG5, JN1 had not been isolated in the country at that time. Currently we have evidence that the vaccine induces a neutralizing response against the most recent Omicron variants, but with a limited number of samples. These studies were carried out at the Helmholtz Center in Germany using pseudoviruses. Since there were a limited number of samples and different experimental conditions, we decided not to include it in the manuscript.
Comments 2: Although the immune response of COVID-19 convalescents increased 14 days after vaccination with both vaccines, it seems that the neutralizing antibody titer against Omicron-BA.5 did not increase much.
Response 2: Although neutralization titers against variant D614G were higher than those against Omicron, both vaccines provided significant increase of neutralization titers against the BA.5 variant. It is important to remark that study participants were infected with COVID-19 before December 2021, the time of the introduction of Omicron variant in the country. While the vaccine is reinforcing the response against the epitopes of the original strain, the individuals included in the study have never been in contact with Omicron, so in the case of Omicron the response is only reinforced against conserved regions, therefore a greater response against the original variants than against Omicron is expected. In our clinical trial, the protocol was established to evaluate neutralization titers in at least 10% of the studied population because the complexity of the neutralization technique against the live virus and the security level required do not allow the evaluation of a high flow of samples such as antibody titers.
Comment 3: The number of samples tested for neutralizing antibodies is far less than that for other test indicators.
Response 3: In our clinical trial, the protocol was established to evaluate neutralization titers in at least 10% of the studied population because the complexity of the neutralization technique against the live virus and the security level required do not allow the evaluation of a high flow of samples such as antibody titers.
Comments 4: What is cellular response look like in the trial? Have the authors ever tested?
Response 4: In the clinical trial protocol, the only variables to be evaluated were the serological ones. However, previous cellular immunological response studies, were assessed during intervention studies with the Abdala vaccine.. ELISPOT experiments showed IFNg secretion levels in vaccinees, comparable to those of the naturally infected subjects. On the other hand, flow cytometry experiments showed increase of T-cells population, CD4+/CD154+/ IFNγ+ and CD8+/ IFNγ+ and CD8+/ TNFα+. The cellular response was maintained over time and increased after a booster dose.
Comments 5: Inadequate detail is provided in the population - Have you ever considered disease classfication under infection status or vaccination history before enrolled in this study.
Response 5: In the manuscript, Table 2 provides the COVID-19 disease category stratification in the study. However, current manuscript only covers the main immunological variables of the study. We also consider important to analyze immunological results according to disease classification. Such analysis is already being prepared for a second publication related to this clinical trial. Per protocol, as an exclusion criterion, individuals that received any vaccine treatment (license or in research) of a coronavirus vaccine were not included. We will add l study exclusion criteria as an appendix to the manuscript.
Reviewer 2 Report
Comments and Suggestions for Authors
The authors analyzed the safety and immunogenicity of the intranasal vaccine candidate Mambisa and the intramuscular vaccine Abdala used as booster doses for COVID-19 convalescents. The booster dose was applied to COVID-19 convalescents to evaluate the capacity of both vaccines to enhance immune response to SARS-CoV-2.
The materials and methods section is described qualitatively. Adverse events after vaccine administration, as well as the immunogenicity of the vaccine, were assessed. The type and intensity of adverse events were classified into several categories. To assess the immunogenicity of each vaccine, anti-RBD antibody titer, inhibition of RBD-ACE2 binding, and neutralizing antibody levels were determined. As a result of the study, the authors showed that both products surpassed the clinical trial’s immunogenicity endpoints. The booster dose with both vaccines strengthened participants’ immune response. I recommend accepting the manuscript for publication in its present form.
Author Response
Thank you very much for reviewing the manuscript
Reviewer 3 Report
Comments and Suggestions for Authors
1. What is the significance of this experiment in COVID-19 convalescents, and why is it necessary? The author did not clearly explain this in the Introduction.
2. Figure 1 should be simplified for better understanding.
(1)"Allocation" can be presented without the extra "N = 30". For example, "Abdala n=30 Intervention n=30" should be changed to "Abdala Intramuscular n=30" and "Mambisa Ayomizer n=30 Intervention n=30" should be changed to "Mambisa Ayomizer n=30".
(2)The set of safety is the set of allocation, known as the full analysis set (FAS), which aims to include subjects as close as possible to the intention-to-treat (ITT) principle. The authors should specify the data set for analysis using FAS, PPS, or other data sets following international terms.
3. In Figure 1B, why use Atomizer for Phase 2? What are the advantages?
4. In Figure 1B, "Discontinued intervention" should not be inside the box of "Intervention" but should be placed above or before this box. Details about this subject should be provided. If this subject is included in the FAS set (n=1041), this case should be included when analyzing the safety set (n=308). If this subject did not participate in the experiment before intervention, it should be excluded before randomization and placed in the "Excluded" category (n=28).
5. The authors should avoid ambiguities in describing the results. For example, stating that “Abdala and Mambisa were well tolerated” (line 266) conflicts with the later statement that “Of the 746 AE reported…related to vaccination, 91.3% (115/126) and 81.1% (503/620) were respectively” (lines 267-269). With more than 80% of AEs related to vaccination, it becomes challenging to convincingly argue that the vaccines are safe.
6. The authors should double-check all numbers in the article. For instance, "AE were reported in 333 participants (51 in Phase 1 and 281 in Phase 2)" (lines 266-267) might actually be “282” in Phase 2. The discrepancy between “290” and “282” in “Vaccination-related AE were reported in 27.9% (290/1041) of participants” (line 287) needs clarification. It is also unclear if it is 333 participants who reported AE or 341 AEs (line 266).
7. When describing severe intensity AE (lines 282-286), it could be clarified that “In this phase, 4.2% (5/120) of participants reported AE of severe intensity, including one with insomnia (Abdala group), one with hypertension and one with pruritus (Mambisa S group), one with headache, sore throat, and otalgia, and one with otalgia and nasal congestion (Mambisa D group).”
8. The sentence "Pain at the injection site was a frequent local AE" (lines 296-298) could be revised to start with “As in Phase 2, besides the above similarities with Phase 1,”.
9. Why weren't neutralizing antibodies against the delta and beta strains tested again in Phase 2? If data is available, the results of Phase 1 could be confirmed.
10. Regarding Figure 3 and Figure 4, it might be more beneficial to show the geometric mean titer (GMT) instead of the median. Additionally, duplicate data in Table 4 and Figures 3 or 4 should be removed from Table 4 to improve readability and understanding.
11. The author should review the format of Table 4, especially regarding the three groups of Mambisa (in Phase 1), to avoid potential confusion.
Author Response
Thank you very much for reviewing the manuscript.
Comments 1: What is the significance of this experiment in COVID-19 convalescents, and why is it necessary? The author did not clearly explain this in the Introduction.
Response 1: The study is necessary because it is widely demonstrated that convalescents are not adequately protected against new SARS-CoV-2 infections and that protection is increased with vaccination, however, most of these reports are with mRNA or live vector vaccines. For greater precision we change the last paragraph of the Introduction:
The objective of this clinical trial was to assess the safety and immunogenicity of Abdala and Mambisa vaccines when administered as booster shots to individuals who have recovered from COVID-19. This is the first report on the immune response of convalescent individuals to SARS-CoV-2 after receiving a booster dose of the Abdala or Mambisa vaccines. Furthermore, it is also the first study to document the use of a nasal subunit vaccine.
Comments 2: Figure 1 should be simplified for better understanding.
(1)"Allocation" can be presented without the extra "N = 30". For example, "Abdala n=30 Intervention n=30" should be changed to "Abdala Intramuscular n=30" and "Mambisa Ayomizer n=30 Intervention n=30" should be changed to "Mambisa Ayomizer n=30".
Response 2.1: We appreciate the suggestion. The manuscript was modified accordingly.
(2)The set of safety is the set of allocation, known as the full analysis set (FAS), which aims to include subjects as close as possible to the intention-to-treat (ITT) principle. The authors should specify the data set for analysis using FAS, PPS, or other data sets following international terms.
Response 2.1: We appreciate the suggestion. The manuscript was modified accordingly.
Comments 3: The atomizer was continued to be employed due to its availability and the fact that this type of device allows for more precise dosing of the vaccine into the atrium or anterior region of the nasal respiratory epithelium. All of the aforementioned details are included in the manuscript.
Response 3: The atomizer was continued to be employed due to its availability and the fact that this type of device allows for more precise dosing of the vaccine into the atrium or anterior region of the nasal respiratory epithelium. All of the aforementioned details are included in the manuscript.
Comments 4: In Figure 1B, "Discontinued intervention" should not be inside the box of "Intervention" but should be placed above or before this box. Details about this subject should be provided. If this subject is included in the FAS set (n=1041), this case should be included when analyzing the safety set (n=308). If this subject did not participate in the experiment before intervention, it should be excluded before randomization and placed in the "Excluded" category (n=28).
Response 4: We appreciate the suggestion. The manuscript was modified accordingly. The subject who experienced an adverse event during the course of the intervention, didn’t received the vaccine, but was incorporated into the FAS for safety purposes. The subject was randomly assigned to Mambisa group.
Comments 5: The authors should avoid ambiguities in describing the results. For example, stating that “Abdala and Mambisa were well tolerated” (line 266) conflicts with the later statement that “Of the 746 AE reported…related to vaccination, 91.3% (115/126) and 81.1% (503/620) were respectively” (lines 267-269). With more than 80% of AEs related to vaccination, it becomes challenging to convincingly argue that the vaccines are safe.
Response 5: The expression was removed from the text.
Comments 6: The authors should double-check all numbers in the article. For instance, "AE were reported in 333 participants (51 in Phase 1 and 281 in Phase 2)" (lines 266-267) might actually be “282” in Phase 2. The discrepancy between “290” and “282” in “Vaccination-related AE were reported in 27.9% (290/1041) of participants” (line 287) needs clarification. It is also unclear if it is 333 participants who reported AE or 341 AEs (line 266).
Response 6: We appreciate the suggestion. Numbers have been corrected.
Comments 7: When describing severe intensity AE (lines 282-286), it could be clarified that “In this phase, 4.2% (5/120) of participants reported AE of severe intensity, including one with insomnia (Abdala group), one with hypertension and one with pruritus (Mambisa S group), one with headache, sore throat, and otalgia, and one with otalgia and nasal congestion (Mambisa D group).”
Response 7: We appreciate the suggestion. The manuscript was modified accordingly.
Comments 8: The sentence "Pain at the injection site was a frequent local AE" (lines 296-298) could be revised to start with “As in Phase 2, besides the above similarities with Phase 1,”.
Response 8: We appreciate the suggestion. The manuscript was modified accordingly.
Comments 9: Why weren't neutralizing antibodies against the delta and beta strains tested again in Phase 2? If data is available, the results of Phase 1 could be confirmed.
Response 9: It would be interesting to confirm the neutralization results against the Beta and Delta strains obtained in Phase I. However, in Phase II, neutralization titers were only evaluated against D614G and the Omicron BA.5 variant. Since the last VOC was already circulating at that time, we decided to focus our efforts and limited BSL3 capacity on evaluating these variants.
Comments 10: Regarding Figure 3 and Figure 4, it might be more beneficial to show the geometric mean titer (GMT) instead of the median. Additionally, duplicate data in Table 4 and Figures 3 or 4 should be removed from Table 4 to improve readability and understanding.
Response 10: We have removed GMT values and Days from Table 4 to make it clearer for reading and understanding.
Comments 11: The author should review the format of Table 4, especially regarding the three groups of Mambisa (in Phase 1), to avoid potential confusion.
Response 11: We have removed several data from the table for better reading and understanding, such as days lines and GMT values..
Round 2
Reviewer 1 Report
Comments and Suggestions for Authors
On page 22, is the data from Phase 1 or Phase 2? Please confirm.
And also, D614G can not be defined as VOC, please modify.
Author Response
Thank you very much for your review.
Comments 1: On page 22, is the data from Phase 1 or Phase 2? Please confirm
Response 1: On page 22 in my word version what you find is the first part of Table 4, which corresponds to Phase I. If this is not the case, could you be more specific in which part of the text you want to clarify to which Phase it belongs?
Comments 2: And also, D614G can not be defined as VOC, please modify.
Response 2: We appreciate the suggestion. The manuscript was modified accordingly.
Reviewer 3 Report
Comments and Suggestions for Authors
GMT should be more suitable than the median for presenting neutralizing antibody results. Therefore, I recommend that only GMT results be shown in Figure 3 and 4, with the numbers and redlines directly.
Author Response
Thank you very much for your review
Comments 1: GMT should be more suitable than the median for presenting neutralizing antibody results. Therefore, I recommend that only GMT results be shown in Figure 3 and 4, with the numbers and redlines directly.
Response 1: We appreciate the suggestion. The manuscript was modified accordingly.